# The Usefulness of Prognostic Tools in Breast Cancer Patients with Brain Metastases

**DOI:** 10.3390/cancers14051099

**Published:** 2022-02-22

**Authors:** Joanna Kufel-Grabowska, Anna Niwińska, Barbara S. Radecka, Shan Ali, Tomasz Mandat, Renata Duchnowska

**Affiliations:** 1Department of Oncology, Poznań University of Medical Sciences, 61-701 Poznań, Poland; joanna.kufel-grabowska@skpp.edu.pl; 2Department of Breast Cancer and Reconstructive Surgery, Maria Sklodowska-Curie National Research Institute of Oncology, 02-781 Warsaw, Poland; anna.niwinska@pib-nio.pl; 3Department of Oncology, Institute of Medical Sciences, University of Opole, 45-040 Opole, Poland; brad@onkologia.opole.pl; 4Neurology Department, Mayo Clinic, Jacksonville, FL 32224, USA; ali.shan@mayo.edu; 5Department of Neurosurgery, Maria Sklodowska-Curie National Research Institute of Oncology, 02-781 Warsaw, Poland; tomasz.mandat@pib-nio.pl; 6Department of Oncology, Military Institute of Medicine, Szaserów St. 128, 04-141 Warsaw, Poland

**Keywords:** brain metastases, breast cancer, prognostic index, nomogram

## Abstract

**Simple Summary:**

Due to the variability of an individual’s prognosis and the variety of treatment options available to breast cancer (BC) patients with brain metastases (BM), establishing the proper therapy is challenging. Since 1997, several prognostic tools for BC patients with BM have been proposed with variable precision in determining the overall survival. The majority of prognostic tools include the performance status, the age at BM diagnosis, the number of BM, the primary tumor phenotype/genotype and the extracranial metastases status as an outcome of systemic therapy efficacy. It is necessary to update the prognostic indices used by physicians as advances in local and systemic treatments develop and change the parameters of survival. Free access to prognostic tools online may increase their routine adoption in clinical practice. Clinical trials on BC patients with BM remains a broad field for the application of prognostic tools.

**Abstract:**

Background: Determining the proper therapy is challenging in breast cancer (BC) patients with brain metastases (BM) due to the variability of an individual’s prognosis and the variety of treatment options available. Several prognostic tools for BC patients with BM have been proposed. Our review summarizes the current knowledge on this topic. Methods: We searched PubMed for prognostic tools concerning BC patients with BM, published from January 1997 (since the Radiation Therapy Oncology Group developed) to December 2021. Our criteria were limited to adults with newly diagnosed BM regardless of the presence or absence of any leptomeningeal metastases. Results: 31 prognostic tools were selected: 13 analyzed mixed cohorts with some BC cases and 18 exclusively analyzed BC prognostic tools. The majority of prognostic tools in BC patients with BM included: the performance status, the age at BM diagnosis, the number of BM (rarely the volume), the primary tumor phenotype/genotype and the extracranial metastasis status as a result of systemic therapy. The prognostic tools differed in their specific cut-off values. Conclusion: Prognostic tools have variable precision in determining the survival of BC patients with BM. Advances in local and systemic treatment significantly affect survival, therefore, it is necessary to update the survival indices used depending on the type and period of treatment.

## 1. Introduction

Brain metastases (BM) are a serious consequence of breast cancer (BC) progression. They occur in up to a half of patients with metastatic human epidermal growth factor receptor 2 (HER2) positive or triple-negative BC and frequently occur in luminal phenotypes with *PIK3CA* alterations [1,2,3,4]. The incidence of BM is increasing; this may be due to improvements in imaging and prolonged survival due to advances in systemic therapy.

Currently, the treatment of BM in BC patients includes neurosurgery, stereotactic radiotherapy (SRS), whole-brain radiotherapy (WBRT) and systemic therapy (chemotherapy, hormonal therapy, targeted therapy and supportive care) [5].

Due to the variability of an individual’s prognosis and the variety of treatment options available to BC patients with BM, establishing the proper therapy is challenging. A patient’s stratification into prognostic classes includes factors related to the patient and his or her disease(s). Beginning in the 1990s, there was a growing interest in developing prognostic tools to provide accurate prognoses, to guide treatment decision making, provide appropriate and cost-effective care and direct clinical trial design. Several prognostic tools in BC patients with BM have been proposed. Our review summarizes the current knowledge on this topic.

## 2. Materials and Methods

The present review compares different prognostic tools based on a systematic literature search on PubMed/MEDLINE. It is limited to adult patients with newly diagnosed BM regardless of the presence or absence of leptomeningeal metastases. The keywords used were “brain metastases/brain metastasis” with “prognostic index” with “breast cancer” or “nomogram” or “prognostic score” and “validation studies” for studies published between January 1997 and December 2021.

## 3. Results

31 prognostic tools were selected: 13 analyzed mixed cohorts with BC cases and 18 were specific to BC. Table 1 chronologically presents prognostic scores developed on mixed groups of patients and Table 2 presents prognostic models constructed on BC-specific groups. The statistically significant prognostic factors included in the 18 BC-specific prognostic tools are presented in Table 3.

### 3.1. General Prognostic Models in BM

In 1997, the Radiation Therapy Oncology Group (RTOG) developed the first prognostic score for BM patients by using a Recursive Partitioning Analysis (RPA) strategy [6]. The study based its analysis on 1200 patients enrolled in 3 RTOG trials conducted from 1979 to 1993 that were investigating WBRT with or without radiosensitizing agents. Breast cancer patients constituted only 12% of the whole group (N = 137). Out of 13 prognostic variables analyzed, 4 were statistically significant: KPS, the age, the status of the primary lesion and evidence of other systemic metastases. Based on these 4 variables, 3 main classes were established, “class I (KPS ≥ 70, primary lesion controlled, age < 65 years and BM only), class II (not class I or III) and class III (KPS < 70) with a median overall survival (OS) of 7.1, 4.2 and 2.3 months,” respectively. In 1999, Lagerwaard et al. [7] developed the Rotterdam Scoring System (RSS) analyzing 1292 patients with BM retrospectively, in whom 213 (16%) were BC patients. Out of the 12 variables, 3 were statistically significant predictors of OS: performance status ECOG scale, clinical response to steroids (good, moderate, little) and extracranial disease/metastases (ECM) status (no, limited, extensive). RSS divided patients into 3 different groups with a median OS of 6.3, 3.4 and 1.3 months.

In 2000, Weltman et al. [8] developed the Score Index for Radiosurgery in BM (SIR) based on 65 patients (only 5 BC patients) treated from 1993 to 1997. Out of the 7 analyzed variables, 5 were included into the SIR: age (≥60 vs. 51–59 vs. ≤50), KPS (≤50 vs. 60–70 vs. >70), ECM, the largest lesion volume (>13 vs. 5–13 vs. <5 cm^3^) and the number of lesions (≥3 vs. 2 vs. 1). The median OS within the 3 SIR groups was 2.9, 7.0 and 31 months.

Lorenzoni et al. [9] proposed the Basic Score for Brain Metastases (BS-BM) based on the outcome of 110 patients treated with radiosurgery from 1999 to 2003. Out of the 10 potential prognostic factors, three built BS-BM: KPS (50–70, 80–100), primary tumor control (yes vs. no) and ECM (yes vs. no). The four prognostic groups were selected with a median survival of 32, 13, 3.3 and 1.9 months.

To predict the survival of BM patients treated with WBRT, Rades et al. [10] created a scoring system by retrospectively analyzing 1085 patients. Four significant factors were identified: KPS (≥70 vs. <70), age (≤60 vs. >60), ECM (yes vs. no) and interval between tumor diagnosis and WBRT and WBRT (≤8, >8 months). Patients were divided into four groups and for each group, survival was compared for a short (5 × 4 Gy) and a longer-course of WBRT (10 × 3 Gy/20 × 2 Gy). The 6-month survival in 4 groups were 6%, 15%, 43% and 76%, respectively. Based on this data, the authors concluded that patients with a better prognosis should receive a longer course of radiation with a lower dose per fraction to avoid neurotoxicity.

Sperduto et al. [11] analyzed 1960 patients from 5 clinical trials from 1979 to 2001 [6]. There were 222 (11%) BC patients with BM. Out of the 7 prognostic variables analyzed, 4 were significant and those were included in the Graded Prognostic Assessment, “KPS (<70 vs. 70–80 vs. 90–100), age (>60 vs. 50–59 vs. <50), ECM (yes vs. no) and number of BM (1 vs. 2–3 vs. >3)”. Each factor was scored (either 0, 0.5 or 1) and the sum (ranging from 0 to 4) was the final prognostic score. They determined four groups with a median OS of 11, 6.9, 3.8 and 2.6 months. This index was compared to the RTOG-RPA, SIR, and BSBM and it was as prognostic as RPA and more prognostic than the other indices.

Rades et al. [12] updated his index by an analysis of a group of 1797 patients after WBRT. The rate of BC patients with BM was not reported in this group. Out of the 7 potential variables, five impacted the OS: KPS (<70 vs. 70 vs. >70), age (≤60 vs. >60 years), ECM (yes vs. no), the interval between the tumor diagnosis and WBRT (≤6 vs. >6 months) and the number of BM (1 vs. 2–3 vs. ≥4). The 6-month survival rates in the 3 groups were 9%, 41% and 78%.

Barnholtz-Sloan et al. [13] developed a nomogram based on 2367 patients from 7 clinical trials. There were 291 (12%) patients with BC in the analyzed group. The nomogram included variables as follows: the primary site and histology, the status of primary disease, ECM, age, KPS, and the number of BM. The authors concluded that the nomogram provided an individual with an estimate of his or her survival and better informed an oncologist on the patient’s prognosis.

In 2012, Yamamoto et al. [14] proposed a sub-classification of RPA class II, originally developed by Gaspar [6], by dividing this class into 3 subclasses. The reason was that in the RPA classification the majority of patients were in class II and clinical factors varied widely within the category. In the group of 3753 patients after radiosurgery without WBRT, 282 (7%) were BC patients. The patients were treated from 1998 to 2008. The new index was the sum of 4 factors: KPS (90–100% vs. 70–80%), the number of BM (solitary vs. more), the primary tumor status (controlled vs. uncontrolled) and ECM (yes vs. no). In the subgroup of BC patients, the median OS in RPA class II-a, II-b and II-c were: 20.4; 11.6 and 5.9 months, respectively.

Lee et al. [15] established a simple and practical scoring system for patients treated with SRS regardless of the type of primary cancer. Based on 311 patients, two significant prognostic variables were selected: KPS (60–80 vs. 90–100) and the primary tumor status (stable vs. progressed). The median OS in the 3 prognostic groups was 4.4, 8.5 and 20 months. Only 48 (15%) of the analyzed patients were BC patients.

In 2021, Sato et al. [16] developed a graded prognostic model for patients surviving 3 years or more, based on 3237 patients with different primary tumors treated with SRS. Seven statistically significant factors were selected: the number of BM (1 vs. 2–4 vs. ≥5), gender, KPS (≥80 vs. <80), the type of primary tumor (breast/lung/gastrointestinal/other), the volume of BM (<10 vs. ≥10 cc), the status of the primary tumor (controlled vs. uncontrolled), and ECM (yes vs. no). Four prognostic groups were created, with a median OS of 6.0, 12.9, 23.5 and 36.3 months.

Zhou et al. [17] presented a new individualized nomogram for predicting survival in patients after SRS, utilizing driver genes and volumetric surrogates. The nomogram integrated four prognostic factors: KPS, gene alterations (*EGRF* and *ALK* in lung cancer), the LDH level (<200 vs. >300 U/L), and the cumulative tumor volume (<3.5 vs. ≥3.5 cm^3^). Three risk groups were selected with a significant difference in OS. In the total group of 356 patients, only 38 (10.7%) were BC patients.

Fan et al. [18] constructed a nomogram predicting the OS based on 230 patients treated with radiation therapy from 2012 to 2016. BC patients counted for 15% (33) of all patients. Apart from 5 commonly used prognostic factors (age, cancer diagnosis, KPS, ECM, number of BM), the new nomogram integrated two new ones: systemic therapy before WBRT and systemic therapy after WBRT. The nomogram predicted 6 and 12-month OS probability after radiotherapy with a concordance index (c-index) of 0.70.

### 3.2. Breast Cancer-Specific Prognostic Models in BM

In 2005, Claude et al. [19] retrospectively reviewed 120 patients with BC and BM treated with WBRT. The prognostic factors (age, performance status, tumor characteristics and pretreatment modalities) were analyzed and two of them had the statistical power to predict the OS: ECOG 0–1 vs. >1 and lymphopenia ≤700 G/L vs. >700 G/L. The OS of patients with good and poor prognoses were 7.0 and 2.0 months.

Le Scodan et al. [20] analyzed 117 BC patients with BM treated with WBRT from 1998 to 2003. Nine prognostic variables were assessed as potential predictors of the OS. Three adverse prognostic factors were selected: RPA class III or KPS < 70, negative hormone receptor (HR) status and lymphocyte count ≤700 G/L. The OS of patients without negative prognostic factors was 15.0 months, 5 months with one adverse prognostic factor and 3 months with more than one prognostic factor.

In 2009, Park et al. [21] retrospectively analyzed 125 patients with BC and BM. Eight prognostic factors were analyzed and only three of them (poor performance status ECOG ≥2, HER2-positivity and no additional systemic therapy) were identified as risk factors for a worse prognosis.

Nieder et al. [22] retrospectively developed a prognostic score on 83 BC patients based on 4 significant variables: KPS (<70 vs. ≥70), ECM (yes vs. no), the number of BM (1 vs. more) and the median time interval from breast cancer diagnosis to metastases (<38 vs. ≥38 months). In the 3-tiered score, the median survival was 16.0, 5.5, and 2.7 months while the 4-tiered score was 16.0, 5.5, 3.6, and 2.7 months, respectively.

In 2012, Sperduto et al. [24] developed Breast Graded Prognostic Assessment (Breast GPA) based on the retrospective analysis of 400 BC patients treated from 1993 to 2010. Out of the five variables analyzed, three significant prognostic factors were selected: KPS, a biological subtype of BC (basal, luminal A, luminal B, HER2) and age (only for patients with KPS 60–80). Four prognostic groups were developed with the median survival time of 3.4, 7.7, 15.0 and 25.0 months. These data confirmed the value of the tumor subtype and the OS.

In a prospective study, Niwińska et al. [23] published a new Breast Cancer Recursive Partitioning Analysis prognostic index based on 441 consecutive patients treated from 2003 to 2009. Three prognostic classes were selected with a median survival of 29.0, 9.0 and 2.4 months. Class I included patients with 1–2 BM, without or controlled ECM and KPS of 100. Class II included patients with multiple BM with KPS < 70. Class II included all other cases. Patients in class I required aggressive treatment, patients in class II required an individual approach and patients in class III required only WBRT or symptomatic therapy.

Ahn et al. [25] published a new breast cancer-specific nomogram developed on a retrospective analysis of 171 BC patients from Korea. It included the following variables: KPS (≥70 vs. <70), the age ≥70 vs. <70), the biological subtype: TNBC (yes vs. no), HER2-positivity (yes vs. no), ECM controlled (complete (CR) or partial response (PR), stable disease (SD)) vs. uncontrolled (progressive disease (PD)) and trastuzumab use (yes vs. no). The four prognostic groups were selected with the median OS from detection of BM of 3.7, 7.8, 10.7 and 19.2 months.

In turn, Marko et al. [26] proposed the nomogram predicting OS in BC patients with BM based on 261 patients treated from 1999 to 2008. Nine factors were included in the nomogram: the dimension of the largest BM, the number of CNS metastases, KPS, the age at the detection of BM, ECM (yes vs. no), the BC stage (I, II, III, IV) and expression of the estrogen receptor (ER) (positive vs. negative), progesterone receptor (PR) (positive vs. negative) and HER2 (positive vs. negative). The c-index was 0.67, which was better than the concordance of RPA, GPA, DS-GPA and modified DS-GPA (range 0.51–0.61).

Le Scodan et al. [27] developed a new prognostic score including molecular subtypes of BC and treatment. The analysis included 130 patients with BC and BM. Statistically significant prognostic variables were: the biological subtype (hormone receptor, HR−/HER2−, HR+/HER2−; HR ± /HER2+), treatment with trastuzumab (yes vs. no), KPS (≥70 vs. 70), age (≥50 vs. <50) and lymphopenia at BM diagnosis (>700 vs. ≤700). Three classes were defined with a median OS of 19.5, 12.5, and 3.5 months.

Rades et al. [28] constructed a simple model to estimate the 6-month survival based on the data of 230 BC patients and BM treated only with WBRT. The two selected prognostic variables were KPS (<70 vs. ≥70) and ECM (yes vs. no). Three groups were defined, with the rate of 6 months survival of 10%, 55% and 78%, respectively.

In 2014, Ahluwalia et al. [29] validated the Breast-GPA (Sperduto 2012) and revealed that it was prognostic for the OS; however, separation between the groups was variable. Based on the analysis of 371 patients, they proposed a Revised Diagnosis-Specific GPA (Revised DG-GPA) by combining Breast-GPA with 3 other factors: number of ECM (0 or 1; >1), the status of the primary tumor (controlled yes vs. no) and leptomeningeal metastases (LEP) (present or absent). Three prognostic groups were defined with a median survival of 4.0, 11.9 and 23.4 months, respectively.

Yang et al. [30] developed the Point Scoring System based on 136 BC patients with 1 to 3 BM who underwent SRS from 2000 to 2012. The number of BM (1 vs. 2–3), ECM (active vs. absent/stable) and the biological subtype (TNBC being the worst) were the factors used to create the index. The median OS in the four prognostic groups were 9.2, 15.6, 25.0 and 45.0 months.

Subbiach et al. [31] validated the breast-GPA on a cohort of 1552 BC patients. Based on the breast-GPA developed by Sperduto in 2012, the authors added the number of BM to the original breast-GPA index, creating a Modified Breast-GPA Index. The median OS of the four prognostic groups were: 2.6, 9.2, 19.9 and 28.8 months.

Huang et al. [32] proposed a new nomogram for predicting survival in BC patients and BM. There were 17% of patients with LEP in the assessed group. Based on the retrospective analysis of 411 patients, the nomogram included 6 significant variables: KPS (70–80 vs. <70), the biological subtype of BC (luminal A, luminal B, triple-negative, HER2 positive), LEP (yes vs. no), number of BM (≤3 vs. >3), ECM controlled (CR, PR, SD) vs. uncontrolled (PD) and disease-free survival (>36 vs. ≤36 months).

Xiong et al. [33] established two new nomograms estimating the individual OS and BC-specific survival, based on the data of 789 BC patients and BM from the Surveillance, Epidemiology, and End Results (SEER) program. There were 6 significant variables constructed in the nomogram: age (18–49, 50–64, ≥65), tumor subtype (luminal A, luminal B, triple-negative, HER2 positive), chemotherapy (yes vs. no)), surgery (yes vs. no), the number of ECM sites (0, 1, 2, 3) and the median household income (high vs. low).

In 2020, Sperduto et al. [34] published updated the Breast Graded Prognostic Assessment (Breast-GPA) analyzing 3 groups of patients treated in different periods of time: cohort A, N = 642, 1985–2007; cohort B, N = 400, 1993–2010; cohort C, N = 2473, 2006–2017. The prognostic factors significant for OS were: age (<60 vs. ≥60), KPS (<60 vs. 70–80 vs. 90–100), tumor subtype (TNBC the worst, ER/PR+ HER2+ the best), the number of BM (1 vs. >1) and ECM (present vs. absent). Four prognostic groups were created with a median survival of 6.0, 12.9, 23.5 and 36.3 months. The authors stated that updated Breast-GPA offered a more accurate method to estimate survival. They concluded that it facilitated clinical decision-making, end-of-life care and appropriate stratification of future clinical trials.

In 2020, Weykamp et al. [35] developed a new prognostic index for radiosurgery of BM in BC patients: the Breast Cancer Stereotactic Radiotherapy Score (bSRS). The variables included in the score were KPS (≤70 vs. 80 vs. 90–100), HER2 status (negative vs. positive) and ECM (yes vs. no). This tool discriminated against 3 prognostic groups with a median OS of 9.0, 16.5 and 46.0 months.

Liu et al. [36] proposed and validated a new nomogram, named NCCBM using a large cohort of 975 BC patients from the SEER database diagnosed from 2011 to 2014. The following variables were selected: age, race, surgery, radiation therapy, chemotherapy, laterality, grade, molecular subtype, and extracranial metastatic sites. The c-index of the NCCBM was 0.69 (95%CI, 0.67 to 0.71) in the training set and 0.70 (95%CI, 0.68 to 0.73) in the validation set. The main limitation of this nomogram was that clinical data on the tumor subtype and distant metastases were collected only after 2010 in the SEER database and therefore limited the sample size of this study. Further, information about disease recurrence or subsequent sites of disease involvement was not collected and therefore this study was unable to investigate patients who developed BM later in their disease course. Moreover, lack of detailed treatment information for patients with BM and KPS does not allow for comparison NCCBM to other prognostic models directly.

### 3.3. External Validation of the Applicability of Different Survival Prognostic Scores

Three separate groups of scientists [20,37,38] confirmed the prognostic value of the RPA score (Gaspar, 1997) in patients with BC and BM. Nieder et al. [22] in 2009 validated RPA, GPA, BSBM, SIR and Rades score on the group of 85 BC patients and revealed that the scores that performed best were RPA and SIR but the c-index was not used. In 2011, Villa et al. [39] validated 3 prognostic indices: RPA, GPA and BS-BM on a group of 285 patients (17% BC patients). Harrell’s c-index values were 0.58, 0.61 and 0.58 for the GPA, BSBM and RPA, respectively. The authors concluded that these indices had a limited long-term prognostication capability.

Ahn et al. [25] in 2012 validated the Breast-GPA on a group of 171 BC patients treated from 2000 to 2008. In this cohort, the Breast-GPA did not discriminate among prognostic classes and the prediction model for the 1-year survival probability (area under the curve (AUC) of 0.55). In turn, Braccini et al. [40] in 2013 compared 7 published prognostic indexes: RPA (Gaspar, 1997), GPA (Sperduto, 2008), BS-BM (Lorenzoni, 2004), Breast-RPA (Niwińska, 2012), Breast-GPA (Sperduto, 2012), Le Scodan’s score (2012) and the clinico-biological score developed in the phase I study (The Phase 1 Prognostic Score, P1PS) [41] in 250 BC patients with BM treated from 1995 to 2010. The analysis revealed that all the indices were able to discriminate patients with statistical significance (*p* < 0.001) for the OS according to the prognostic category. Pairwise comparisons of each prognostic index revealed statistically significant differences in survival between prognostic classes, except for the breast-GPA classes I vs. II, BS-BM scores 1 vs. 2 and Le Scodan’s scores I vs. II. There were no significant differences between all prognostic indices concerning the survival predicting ability. Only minor differences were seen using Harrell’s c-index (range 0.60–0.68). The authors concluded that RPA seemed to be the most useful score. RPA performed better than new prognostic indices because it was the most accurate in identifying patients with long (>12 months) and short (<3 months) survival.

Tabouret et al. [42] in 2014 evaluated the prognostic value and validity of the 6 scoring systems (Sperduto [24,43], Niwińska [23], Nieder [22], Park [21], Le Scodan [20] and Claude [19]) in an independent population of 152 BC patients and BM treated from 1995 to 2011. All scores showed significant prognostic value for the OS and discriminative ability.

In 2015, Subbiah et al. [31] validated the Breast-GPA (Sperduto, 2012) on a cohort of 1552 BC patients with BM. Based on the Breast-GPA developed by Sperduto, the authors added the number of BM to the original Breast-GPA index, creating a Modified Breast-GPA Index. The c-index for original Breast-GPA was 0.78 and for the Modified Breast GPA it was 0.84, showing that the latter had a better performance in terms of discrimination.

Castaneda et al. [44] in 2015 validated RPA and Breast-GPA indices on a group of 215 BC patients with BM. The authors stated that both indices were useful in clinical practice. However, the addition of ECM status to the Breast-GPA score improved the validity of this index.

Further, Laakmann et al. [45] compared 9 prognostic scores in 139 BC patients and BM treated with radiation therapy between 2007 and 2012. The prognostic value and accuracy of RPA, RPA II, GPA, BS-BM, Breast-GPA, Breast-RPA, Rades Score 2011, Germany Score and Breast Rades Score were assessed. The analyses revealed that the majority of the scores were associated with the OS, but the GPA was the most accurate at identifying patients with a survival of more than 1 year and Breast-GPA was the best in selecting patients with a survival of less than 3 months.

Huang et al. [32] in 2018 performed an external validation of the RPA, GPA and Breast-GPA on a group of 411 patients. The analysis revealed overlapping between groups I and II in RPA, and between groups II, III and IV in GPA, and unsatisfactory discrimination between groups II and III in Breast-GPA. It was concluded that in those three prognostic models, the value of differentiating a patient’s survival was not satisfactory. The c-index was 0.64, 0.61 and 0.63 for the nomogram Breast-GPA, GPA and RPA, respectively.

Znidaric et al. [46] in 2019 validated the applicability of prognostic models based on 423 BC patients, treated from 2005 to 2015. The analyzed prognostic scores were: Breast-RPA, Breast-GPA, MB-GPA and Simple Survival score for patients with Brain Metastases (SS-BM). All four prognostic classifications proved to be valuable prognostic tools in predicting survival with a c-index range of 0.71–0.768.

Similarly, in 2019 Zhuang et al. [47] undertook a validation of Breast-GPA (Sperduto, 2012) and Modified Breast-GPA (Subbiah, 2015) on the Asian cohort of 282 BC patients and BM diagnosed from 2006 to 2017. Both indices demonstrated moderated discriminative capabilities for the OS. The c-index for GPA and Breast-Modified Breast-GPA was 0.64 and 0.65, respectively. The authors concluded that the inclusion of ECM status could improve its prognostic value.

Lee et al. [15] in 2020 proposed a prognostic index for BM (PIBM) and performed a validation of 4 indices (RPA, SIR, BSBM, GPA) on a group of 311 BC patients treated with gamma knife radiosurgery. All four indices were comparable in regards to the prognostic ability (AUC range 0.59–0.63) and PIBM had the highest prognostic power (AUC~0.66).

Weycamp et al. [35] in 2020 validated 9 prognostic indices (RPA [6], Modified RPA [14], GPA [11], Breast-GPA [24], Modified GPA [31], Updated Breast GPA [34], SIR [8], BS-BM [9] and Point Scoring System [30]) for SRS in 95 BC patients with BM treated from 2005 to 2016. Only two of them showed a significant c-index: Breast GPA (0.63) and Modified Breast-GPA (0.66).

## 4. Discussion

Since the first Recursive Partitioning Analysis published in 1997, a series of prognostic tools have been developed in BC patients with newly diagnosed BM to facilitate clinical decision-making and appropriate stratification to local and systemic therapy. This review of prognostic tools illustrates how over time, progress in the understanding of the biology of BC and new effective systemic treatments have influenced the prognosis of patients with BM [6,7,8,9,10,11,12,13,14,15,16,17,18,19,20,21,22,23,24,25,26,27,28,29,30,31,32,33,34,35].

The majority of prognostic tools in BC patients with BM include the performance status, the age at BM diagnosis, the number of BM (rarely volume), the primary tumor phenotype/genotype and the ECM status as an outcome of the systemic therapy efficacy (Table 3) [19,20,21,22,23,24,25,26,27,28,29,30,31,32,33,34,35]. Although the results of several analyses agree that patients with better performance status achieve a longer survival, several researchers emphasize that this criterion carries a risk of bias because the clinical assessment of performance status is subjective. The most often used scale to assess the performance status was Karnofsky’s scale with a cut-off point of 70 (i.e., the patient could not work, but did not require care) (Table 2 and Table 3). Some studies described 3 prognostic subgroups: patients in a very good general condition (90–100 points), patients in a poor general condition (≤60 points, or <70 points) and the intermediate group (61–90 points or 70–89 points). Similar controversies concern age because the biological age often differs from the chronological age. The most often used cut-off for age in prognostic tools was 60 or 65 years, less often 50 (Table 2 and Table 3). Also, the prognostic significance of the number of BM has not been clearly defined. Currently, when dealing with 1 to 10 BMs for treatment qualification, BM volume and lesion location seem to be more important than the number alone [5,48,49].

On the other hand, the BC phenotype or genotype is a main predictive factor of systemic therapy efficacy and ECM control. However, studies assessing ECM status as a prognostic factor used various definitions. Typically, it was the presence of distant metastases outside the brain or their absence [22,26,28,30,34,35]. Others analyzed the ECM site (e.g., bone metastases vs. metastases to parenchymal organs or metastases in the liver vs. metastases in other sites) as well as the number of organs outside the brain with metastases [6,33]. ECM was also analyzed in terms of its activity and classified as controlled or uncontrolled [23,25,32]. Furthermore, the disease progression was defined as the appearance of new lesions or the progression of previous lesions found in computed tomography 3 months before radiotherapy or if the examination was not performed, one month after irradiation. Still, uncontrolled ECM in BC patients was associated with a worse prognosis of patients and a shorter overall survival [22,23,25,26,27,28,30,32,33,34,35].

Advanced TNBC and HER2-positive BC have a higher risk of BM [2,3]. It is a natural course of advanced disease. On the other hand, an effective anti-HER2 therapy allows one to extend the time until symptomatic BM. An exploratory analysis from a phase III trial CLEOPATRA suggests that pertuzumab, trastuzumab, and docetaxel in a one line setting in HER2-positive advanced breast cancer delays the onset of CNS [50]. Similarly, in the NEfERT-T study, neratinib-paclitaxel delayed the onset and reduced the frequency of central nervous system progression [51]. Moreover, an effective anti-HER2 treatment after BM improved the OS, and new drugs that penetrate the blood-brain barrier were active in the brain and in the ECM [52]. According to the literature, the median OS in BC patients with BM is the longest in patients with HER2-positive treated with anti-HER2 regiments, then with luminal HER-2-negative and the shortest in TNBC [53,54]. Other prognostic factors included in prognostic tools, e.g., the time of diagnosis to the occurrence of BM, or disturbances in laboratory parameters (lymphopenia, increase in lactate dehydrogenase) seem to be derivative of the disease control [17,19,20,27].

Predicting the survival of BC patients with BM is difficult; therefore, prognostic tools are crucial in stratifying different patients’ outcomes. However, they are limited by their retrospective nature and may underestimate survival in the modern era with the growing number of effective systemic agents. Furthermore, the prognostic tool developed for single institution cohorts might be biased by institutional practice patterns; therefore, external validation for new prognostic tools are the gold standard and should be obtained whenever possible [55,56]. Moreover, this validation should be performed in a cohort of patients with similar characteristics and demographics, and treated during a similar time period. The diversity of populations between cohorts may explain discrepancies in results and may reveal that they are not as predictive for the OS as the original one and may even underestimate the real survival [57].

International multidisciplinary recommendations, EANO-ESMO and NCCN, do not recommend prognostic tools in BC patients with BM [5,49]. The eligibility criteria in clinical trials with new systemic therapies in these patients are based mainly on the performance status, stabilization after local treatment (surgery and/or WBRT), the need for immediate local therapy (in patients with untreated BM) and the need for a daily dose of corticosteroids to control symptoms of BM. Furthermore, patients with leptomeningeal disease are usually excluded from clinical studies dedicated to BM (www.clinicaltrials.gov; accessed on 25 January 2022).

In clinical practice, the decision on the treatment sequence of BC patients with newly diagnosed BM should be made by a multidisciplinary team including a medical oncologist, radiotherapist and neurosurgeon, according to the accepted standard operating procedures (SOP). In the management algorithm, it is crucial to provide supportive care to patients with a poor prognosis and an expected survival of less than 3 months; for example, patients with uncontrolled ECM and/or more than 10 BM where systemic pharmacotherapy and WBRT are mainly used. In turn, for the minority of patients, with small and few lesions (1 to 10) depending on the volume (<15 mL), who may experience long-term survival or even cure, several approaches are used in combination (surgery followed by SRS/SRT or systemic pharmacotherapy, SRS/SRT or systemic pharmacotherapy) [5,49].

The universal, ideal prognostic tool should be simple and easily usable. Electronic access to such indices improves their usefulness in clinical practice, e.g., for the modified updated Breast GPA index (a free online calculator available at brainmetgpa.com). Furthermore, new prognostic tools in BC patients with BM should be used more in clinical trials.

## 5. Conclusions

Prognostic tools have variable precision in determining the survival of BC patients with BM. Progress in local and systemic treatment significantly affects the parameters of survival. Hence, it is necessary to update the prognostic indices used, depending on the period of treatment. Free access to prognostic tools online may increase the frequency of their use in clinical practice. Clinical trials in BC patients with BM remain a broad field for the application of prognostic tools.

## Figures and Tables

**Table 1 cancers-14-01099-t001:** Prognostic tools developed on mixed groups of patients with BM (from year 1997 to 2021).

Author, Prognostic Tool, Year of Publication	N (Total and BC)	Prognostic Factors Included into Index	Prognostic Class/Groups and OS (m)
Gaspar [6], Recursive Partitioning Analysis (RPA), 1997	1200/137	KPS (≥70 vs. <70), age (≥65 vs. <65), primary lesion status (controlled vs. uncontrolled), ECM (single vs. multiple)	I: 7.1 II: 4.2 III: 2.3
Lagerwaard [7], The Rotterdam Scoring System, 1999	1292/213	ECOG performance status (0–3), clinical response to steroids (good, moderate, little), ECM (no vs. limited vs. extensive)	I: 6.3 II: 3.4 III: 1.3
Weltman [8], Score Index for Radiosurgery (SIR), 2000	65/5	Age (≥60 vs. 51–59 vs. ≤50), KPS (≤50 vs. 60–70 vs. >70), ECM status (NED/CR vs. PR/SD vs. PD), largest lesion volume (>13 vs. 5–13 vs. <5 cm^3^) and number of lesions (≥3 vs. 2 vs. 1)	I: 2.9 II: 7.0 III: 31.0
Lorenzoni [9], Basic Score for Brain Metastases (BS-BM), 2004	110/20	KPS (50–70, 80–100), primary tumor control (yes vs. no), ECM status (yes vs. no)	I: 32.0 II: 13.0 III: 3.3 IV: 1.9
Rades [10], Rades score, 2008	1085/207	KPS (≥70 vs. <70), age (≤60 vs. >60), ECM (no vs. yes) and interval between tumor diagnosis and WBRT (≤6 vs. >6 months)	6-m. OS I: 76% II: 43% III: 15% IV: 6%
Sperduto [11], Graded Prognostic Assessment (GPA), 2008	1960/222	KPS (<70 vs. 70–80 vs. 90–100), age (<50 vs. 50–59 vs. >60), number of BM (1 vs. 2–3 vs. >3), ECM status (no vs. yes)	I: 2.6 II: 3.8 III: 6.9 IV: 11.0
Rades [12], Rades score (updated), 2011	1797/UN	KPS (≥70 vs. <70), age (≤60 vs. >60), ECM status (no vs. yes) and interval between tumor diagnosis and WBRT (≤6 vs. >6 months), number of BM (1 vs. 2–3 vs. ≥4)	6-m. OS A: 9% B: 41% C: 78%
Barnholtz-Sloan [13], Nomogram, 2012	2367/291	Primary site and histology (breast and adenocarcinoma, breast and other, lung and adenocarcinoma, lung and large cell, lung and other, lung and small cell, lung and squamous cell, other and adenocarcinoma, GI and other, renal and other, squamous cell and other), status of primary disease (controlled vs. uncontrolled), ECM (present vs. absent), age, KPS (≥70 vs. <70), number of BM (single vs. multiple)	Individual estimate of OS, no prognostic groups
Yamamoto [14], RPA sub-classification, 2012	3753/282	Class II: KPS (90–100% vs. 70–80%), number of BM (solitary vs. more), primary tumor status (controlled vs. uncontrolled), ECM (yes vs. no)	RPA Class II: Breast IIa: 20.4 IIb: 11.6 IIc: 5.9
Lee [15], Prognostic Index for Brain Metastases (PIBM), 2019	311/48	KPS (60–80 vs. 90–100), primary tumor status (stable vs. progressed)	I: 4.4 II: 8.5 III: 20.0
Sato [16], Graded Prognostic Model (GPM ≥ 3 Ys), 2021	3237/370	Number of BM (1 vs. 2–4 vs. ≥5), gender (female/male), KPS (≥80 vs. <80), type of primary (breast/lung/gastrointestinal tract/other), volume of BM (<10 vs. ≥10 cc) status of primary cancer (controlled vs. uncontrolled), ECM (yes vs. no)	I: 3.6 II: 6.8 III: 15.8 IV: 32.8
Zhou [17], Nomogram, 2021	356/38	KPS (≥70 vs. <70), gene mutation (*EGFR*, *ALK* in NSCLC), LDH level (<200 vs. >300 U/L), cumulative tumor volume (<3.5 vs. ≥3.5 cm^3^)	Estimate 12 and 24-m. OS; Risk groups: high, medium, low
Fan [18], Nomogram, 2021	230/33	Age, cancer diagnosis (lung adenocarcinoma, breast, kidney, other, lung non-adenocarcinoma, small cell lung cancer, melanoma, gastrointestinal), KPS (≤70, 80, 90–100), ECM (controlled vs. uncontrolled), number of BM (single, multiple), systemic therapy before WBRT and systemic therapy after WBRT	Estimate 6 and 12-m. OS

N, number of patients; BC, breast cancer patients; BM, brain metastases; ECM, extracranial metastases; KPS, Karnofsky Performance Status; ECOG, Eastern Cooperative Oncology Group; WBRT, whole brain radiotherapy; NSCLC, non-small-cell lung cancer; EGFR, epidermal growth factor receptor, ALK, anaplastic lymphoma kinase; UN, unknown; m, months; NED, no evidence of disease; PD, progressive disease; PR, partial remission; SD, stable disease; CR, complete clinical remission.

**Table 2 cancers-14-01099-t002:** Prognostic tools developed on breast cancer patients with brain metastases (from year 2005 to 2021).

Author, Prognostic Tool, Year of Publication	N	Prognostic Factors Included into Tool	Prognostic Groups and Survival (Months)
Claude [19], Claude predictive model for OS, 2005	120	PS ECOG (0–1 vs. >1), lymphocyte count (700 > vs. ≤700 G/L)	I: 7.0 II: 2.0
Le Scodan [20], Le Scodan score, 2007	117	Hormone receptor status (negative vs. positive), lymphocyte count >700 vs. ≤700 G/L, KPS (<70 vs. ≥70), RTOG-RPA	I: 15.0 II: 5.0 III: 3.0
Park [21], Prognostic factors, 2009	125	PS ECOG (>0–1 vs. ≥2), HER2-positivity (yes vs. no), additional systemic treatment (yes vs. no)	I: 49.0 II: 10.6 III: 4.4 IV: 2.2
Nieder [22], Nieder Score, 2009	83	KPS (<70 vs. ≥70), ECM (yes vs. no), number of BM (1 vs. more), median time interval from breast cancer diagnosis to BM (<38 vs. ≥38 months)	I: 16.0 II: 5.5 III: 3.6 IV: 2.7
Sperduto [23], Breast-Graded Prognostic Assessment (Breast- GPA), 2012	400	KPS (50, 60, 70–80, 90–100), biological subtype (basal, luminal A, luminal B, HER2), age (only for patients with KPS 60–80)	I: 3.4 II: 7.7 III: 15.0 IV: 25.0
Niwińska [24], Breast Cancer Recursive Partitioning Analysis (Breast RPA), 2012	441	KPS (100 vs. ≤60), number of BM (1–2 vs. >2), ECM status (no/controlled vs. uncontrolled)	I: 29.0 II: 9.0 III: 2.4
Ahn [25], BC-specific nomogram, 2012	171	KPS (≥70 vs. <70), age ≥ 70 vs. <70), biological subtype: TNBC (yes vs. no), HER2 positivity (yes vs. no), ECM status (CR, PR, SD vs. PD), trastuzumab use (yes vs. no)	I: 3.7 II: 7.8 III: 10.7 IV: 19.2
Marko [26], Nomogram, 2012	261	Dimension of the largest BM, number of CNS metastases, KPS (30–100), age at BC BM, ECM status (non-CNS Mets no vs. yes), expression (positive vs. negative): ER, PR, HER2 and BC stage (I, II, III, IV)	1-year survival 3-year survival 5-year survival
Le Scodan [27], Le Scodan score (updated), 2012	130	KPS (<70 vs. ≥70), age (<50 vs. ≥50), lymphopenia at BM diagnosis (>700 vs. ≤700), biological subtype (HR−/HER2−; HR+/HER2−; HR ± /HER2+), treatment with trastuzumab (yes vs. no)	I: 19.5 II: 12.5 III: 3.5
Rades [28], Simple Survival Score, 2013	230	KPS (<70 vs. ≥70), ECM (yes vs. no)	6-month survival I: 10% II: 55% III: 78%
Ahluwalia [29], Revised Diagnosis-Specific GPA (Revised Breast-GPA), 2014	371	KPS (50, 60, 70–80, 90–100), biological subtype (basal, luminal A, luminal B, HER2), age (only for patients with KPS 60–80), number of ECM (0–1; >1), status of the primary tumor control (yes vs. no), and LEP (present vs. absent)	I: 4.3 II: 11.9 III: 23.4
Point Scoring System [30], 2014	136	Number of BM (1 vs. 2–3), ECM (active vs. absent/stable), biological subtype (TNBC the worst)	I: 9.2 II: 15.6 III: 25.0 IV: 45.0
Subbiah [31], Modified Breast-GPA Index (MB-GPA), 2015	1552	KPS (≤50 vs. 60 vs. 70–80 vs. 90–100), biological subtype (TNBC vs. ER+/HER2− vs. ER−/HER2+ vs. ER+/HER2+), age (≤50 vs. >50), number of BM (1–3 vs. >3)	I: 2.6 II: 9.2 III: 19.9 IV: 28.8
Huang [32], Nomogram, 2018	411	KPS (70–80 vs. <70), biological subtype of breast cancer (luminal A, luminal B, triple negative, HER2), LEP (yes vs. no), number of BM (≤3 vs. >3), ECM (CR, PR, SD) vs. uncontrolled (PD) and disease-free survival (>36 vs. ≤36 months)	1-year survival 2-year survival
Xiong [33], Nomogram, 2019	789	Age (18–49, 50–64, ≥65), tumor subtype (luminal A, luminal B, triple negative, HER2 positive), chemotherapy (yes vs. no)), surgery (yes vs. no), the number of ECM sites (0, 1, 2, 3) and the median household income (high vs. low)	6-months survival 1-year survival 2-year survival
Sperduto [34], Updated Breast Graded Prognostic Assessment (updated Breast- GPA), 2020	2473	Age (<60 vs. ≥60), KPS (<60 vs. 70–80 vs. 90–100), tumor subtype (TNBC the worst, ER/PR+ HER2+ the best), number of BM (1 vs. >1), ECM (present vs. absent)	I: 6.0 II: 12.9 III: 23.5 IV: 36.3
Weykamp [35], Breast Cancer Stereotactic Radiotherapy Score (bSRS), 2020	95	KPS (≤70 vs. 80 vs. 90–100), HER2-receptor expression (negative vs. positive) and ECM control (yes vs. no)	I: 9.0 II: 16.5 III: 46.0
Liu [36], NCCBM, 2021	975	Age (<40, 40–49, 50–59, 60–69, 70–79, ≥80), race (White, Black, Hispanic, Asian/Pacific, Islander, other), surgery (performed, UNK, not performed), laterality (bilateral, UNK, right, left), grade (I, II, III, IV, unknown), subtype (HR+/HER2−; HR+/HER2+; HR−/HER2+; HR−/HER2−; UNK), ECM (UNK, 0, 1, 2, 3)	6-month survival 1-year survival 2-year survival

N, number of patients; BC, breast cancer patients; BM, brain metastases; ECM, extracranial metastases; CR, complete remission, PR, partial remission; SD, stable disease vs. PD, progressive disease; ER, estrogen receptor; PR, progesterone receptor; HER2, human epidermal growth factor receptor type 2; HR, hormone receptor; LEP, leptomeningeal metastasis; TNBC, triple negative breast cancer; ER, estrogen receptor; PR, progesterone receptor; HER2, human epidermal growth factor receptor type 2; KPS, Karnofsky Performance Status; ECOG, Eastern Cooperative Oncology Group; RTOG-RPA, Radiation Therapy Oncology Group Recursive Partitioning Analysis; UNK, unknown.

**Table 3 cancers-14-01099-t003:** Factors related to OS in prognostic tools, in BC patients with brain metastases published from 1997 to 2021.

Authors	PS	Age	ECM	NBM	CS	ER	PR	HER2	BS	STH	LC	IT	DLBM	LEP	PTC	S	MHI	Other
Claude et al. [19]	ECOG										+							RTOG-RPA
Le Scodan et al. [20]	KPS					+	+				+							
Park et al. [21]	ECOG							+		+								
Nieder et al. [22]	KPS		+	+								+						
Sperduto et al. [23]	KPS	+ ^1^							+									
Niwińska et al. [24]	KPS		+	+														
Ahn et al. [25]	KPS	+	+			+	+	+	+	T			+					
Marko et al. [26]	KPS	+	+	+	+	+	+	+					+					
Le Scodan et al. [27]	KPS	+							+	T				+				
Rades et al. [28]	KPS		+															
Ahluwalia MS et al. [29]	KPS	+ ^1^	+						+					+	+			
Yang et al. [30]			+	+					+									
Subbiach et al. [31]	KPS	+		+					+									
Huang et al. [32]	KPS		+	+					+			+		+				
Xiong et al. [33]		+	+ ^2^						+	CHT						+	+	
Sperduto et al. [34]	KPS	+	+	+					+									
Weykamp et al. [35]	KPS		+					+										
Liu et al. [36]		+	+			+	+	+	+	CHT						+		Race, laterality, grade

PS, performance status; KPS, Karnofsky Performance Status; ECOG, Eastern Cooperative Oncology Group; ^1^ only for patients with KPS 60–80; ^2^ sites number; NB, number of brain metastases; ECM, extracranial metastases; BC, breast cancer; CS, clinical stage; ER, estrogen receptor; PR, progesterone receptor; HER2, human epidermal growth factor receptor type 2; BS, biological subtype; LC, lymphocyte count; IT, interval time from breast cancer diagnosis and brain metastases; STH, systemic therapy (yes vs. no); DLBM, Dimension of the largest BM; CHT, chemotherapy; T, trastuzumab; LEP, leptomeningeal metastases (yes vs. no); PTC, primary tumor control (yes vs. no); S, surgery (yes vs. no); MHI, Median household income (high vs. low); RTOG-RPA, Radiation Therapy Oncology Group Recursive Partitioning Analysis.

## Data Availability

The data presented in this study are available in this article.

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
