# Peer review of "The Usefulness of Prognostic Tools in Breast Cancer Patients with Brain Metastases"

_cancers, 2022, doi:10.3390/cancers14051099_

Round 1

Reviewer 1 Report

This manuscript represents a detailed overview of currently developed prognostic tools which allow to stratify breast cancer patients with brain metastasis, providing a score for each patient, on the basis of both clinical and histopathological data. The authors also compare the presented models to define patients’ categories. The fairness of each stratification is measured in terms of median survival times, thus yielding a proximity for risk estimations.
Overall, it is an interesting study, and it is appropriate for being published on this journal.
However, I believe the manuscript needs some minor revisions.
Point 1: The manuscript was submitted by authors as an ARTICLE type. However, as they declare in the Materials and Methods section with the statement “in the present review”, this work is more assimilable to a REVIEW
Point 2: Since in the Introduction a disclosure about breast cancer metastasis is missing, I recommend the authors to improve the manuscript's background including the following work: Breast cancer metastasis. CANCER GENOMICS & PROTEOMICS.
Point 3: The authors should improve the Introduction also mentioning other currently developed prognostic tools for the clinical practice in breast cancer. The following paper provides an example of such a clinical support tool: A Clinical Decision Support System for Predicting Invasive Breast Cancer Recurrence: Preliminary Results. FRONTIERS 2021
Point 4: Likewise, in the Introduction an exhaustive summary of common treatments for brain metastasis in breast cancer patients is required. For this purpose, the author should cite the following work: Treatment strategies for breast cancer brain metastases. BRITISH JOURNAL OF CANCER
Point 5: Finally, the authors should deal with the potentiality of imaging techniques in monitoring breast cancer brain metastasis. An example is provided by the following manuscript: Breast Cancer Brain Metastasis: The Potential Role of MRI Beyond Current Clinical Applications. CANCER MANAGEMENT AND RESEARCH 2020.
Furthermore, these kinds of clinical supports tools are investigated to face other questions related to breast cancer, such as the prediction of both recurrence and therapy response in patients undergoing neoadjuvant chemotherapy (Early prediction of neoadjuvant chemotherapy response by exploiting a transfer learning approach on breast DCE-MRIs. SCIENTIFIC REPORTS 2021; Early Prediction of Breast Cancer Recurrence for Patients Treated with Neoadjuvant Chemotherapy: A Transfer Learning Approach on DCE-MRIs. CANCERS 2021).

Author Response

We thank the Reviewer for the appraisal of our work. Thank you for your attention regarding the type of article, indeed it is more of a review. The Reviewer is correct that we did not include the information on the treatment of brain metastases or the tools for predicting disease recurrence because the article focuses on prognostic tools for overall survival in patients with brain metastasis already present. The paper is for the Special Issue where other papers will deal with the treatment of brain metastases. We do not want to duplicate this information.

Reviewer 2 Report

Thank you for the opportunity to review this manuscript. Here, the authors provide a scoping review on the usefulness of predictive tools in breast cancer patients with brain metastases. 

They conclude heterogeneity of the scores in predicting survival of breast cancer patients. In light of advances in systemic and local treatments, the authors advocate updating the survival indices.

Minor changes: -Barnholtz-Sloan [13], Nomogram, 2012:  adenocarcinoma instead of adenocarcionoma -line 82 leptomeningeal instead of leptomenigeal metastasis -line 86 metastases instead of metastese -line 315 pairwise instead of pair wise  

Author Response

We thank the reviewer for the positive appraisal of our work.